



# Towards a Chemical Mechanism of the Oxidation of Aqueous Sulfur Dioxide via Isoprene Hydroxyl Hydroperoxides (ISOPOOH)

Eleni Dovrou[1,5], Kelvin H. Bates[2], Jean C. Rivera-Rios[3,6], Joshua L. Cox[3], Joshua D. Shutter[3], and Frank N. Keutsch[1,3,4]

[1]John A. Paulson School of Engineering and Applied Sciences, Harvard University, Cambridge, MA 02138, USA.
[2]Harvard University Center for the Environment, Cambridge, MA 02138, USA.
[3]Department of Chemistry and Chemical Biology, Harvard University, Cambridge, MA 02138, USA.
[4]Department of Earth and Planetary Sciences, Harvard University, Cambridge, MA 02138, USA.
[5]Now at Multiphase Chemistry Department, Max Planck Institute for Chemistry, Mainz 55128, Germany.
[6]Now at School of Chemical & Biomolecular Engineering, Georgia Institute of Technology, Atlanta, GA 30332, USA.

*Correspondence to*: Eleni Dovrou (dovrouel@gmail.com) and Frank N. Keutsch (keutsch@seas.harvard.edu)

**Abstract.** In-cloud chemistry has important ramifications for atmospheric particulate matter formation and gas-phase chemistry. Recent work has shown that, like hydrogen peroxide ($H_2O_2$), the two main isomers of isoprene hydroxyl hydroperoxide (ISOPOOH) oxidize sulfur dioxide dissolved in cloud droplets ($SO_{2,aq}$) to sulfate. The work revealed that the pathway of $SO_{2,aq}$ oxidation with ISOPOOH differs from that of $H_2O_2$. We investigate the chemical mechanisms of oxidation of $SO_{2,aq}$ with ISOPOOH in the cloud-relevant pH range of 3-6 and compare them with the previously reported mechanisms of oxidation of $SO_{2,aq}$ with $H_2O_2$, methyl hydroperoxide and peroxyacetic acid. The organic products of the reaction are identified and two pathways are proposed. For 1,2-ISOPOOH, a higher yield pathway via proposed radical intermediates yields methyl vinyl ketone (MVK) and formaldehyde, which can react to hydroxymethanesulfonate (HMS) when $SO_{2,aq}$ is present. A lower yield non-fragmentation oxygen addition pathway is proposed that results in formation of isoprene-derived diols (ISOPOH). Based on global simulations, this mechanism is not a significant pathway for formation of MVK and formaldehyde relative to their gas-phase formation but, as previously reported, it can be regionally important for sulfate production. The study adds to previous work that highlights similarities and differences between gas-phase and cloud-droplet processing of reactive organic carbon.

## 1 Introduction

Isoprene ($C_5H_8$) is the main non-methane biogenic volatile organic compound emitted to the atmosphere with global emission estimates of ~470 Tg C (Guenther et al., 2006; Guenther et al., 2012; St. Clair et al., 2016). In the atmosphere, $C_5H_8$ primarily reacts with hydroxyl radicals (OH) forming peroxy radicals ($RO_2$) after oxygen addition (Wennberg et al., 2018). Under pristine, $HO_2$-dominated (low-NO) conditions, isoprene $RO_2$ react with hydroperoxyl radicals ($HO_2$) to form multifunctional



organic hydroperoxides, isoprene hydroxyl hydroperoxides (ISOPOOH, $C_5H_{10}O_3$), of which 1-hydroxyl-2-hydroperoxyl- and 4-hydroxyl-3-hydroperoxyl-ISOPOOH (1,2-ISOPOOH and 4,3-ISOPOOH, respectively) are the most abundant isomers (Rivera-Rios et al., 2014; Krechmer et al., 2015; St. Clair et al., 2016). ISOPOOH mixing ratios of up to 1 ppb and up to ~2 ppb have been reported in the Amazon rainforest and the Blodgett Forest Research Station in California, respectively (Worton et al., 2013; Liu et al., 2016).

In the gas-phase, ISOPOOH is oxidized by OH to primarily produce isoprene epoxydiols (IEPOX), which contribute to secondary organic aerosol (SOA) mass (Paulot et al., 2009; Surratt et al., 2010; Lin et al., 2012; Nguyen et al., 2014; McNeill, 2015; Zhang et al., 2018). St Clair et al. (2016) investigated the gas-phase oxidation mechanism of ISOPOOH and elucidated some non-IEPOX oxidation pathways, and Krechmer et al. (2015) observed the formation of non-IEPOX low volatile organic compounds that contribute to SOA. These pathways are summarized and compiled into a complete gas phase mechanism in the work of Wennberg et al. (2018).

Having Henry's law constants on the order of $10^5 \, \mathrm{M \cdot atm^{-1}}$, ISOPOOH isomers can partition into cloud and fog water and participate in condensed-phase reactions (Rivera-Rios, 2018). Dovrou et al. (2019b) showed that ISOPOOH oxidizes sulfur dioxide ($SO_{2,aq}$) in cloud and fog water, producing sulfate ($SO_4^{2-}$). Aqueous hydrogen peroxide ($H_2O_2$) oxidizes $SO_{2,aq}$ to sulfate with a 100% yield with a rate constant that increases with decreasing pH (Lind et al., 1987). Dovrou et al. (2019b) reported that at pH=5.5 the rate constant for oxidation of $SO_{2,aq}$ by 1,2-ISOPOOH is equal, within uncertainty, to that of oxidation of $SO_{2,aq}$ by $H_2O_2$ at pH=5.5, while the rate constant of the oxidation for $SO_{2,aq}$ by 4,3-ISOPOOH is an order of magnitude lower. The rate constants for the reactions of both ISOPOOH isomers with $SO_{2,aq}$ have a much smaller pH dependence than that of $H_2O_2 + SO_{2,aq}$. Similar to $H_2O_2$, the sulfate yield at pH=4.5 and 3 is 100% for both ISOPOOH isomers. However, Dovrou et al. (2019b) showed that for pH=5.5 the sulfate yield is only 67% and 83% of the reacted $SO_{2,aq}$ for 1,2-ISOPOOH and 4,3-ISOPOOH, respectively (Table 1). Due to its higher abundance, solubility and rate of reaction with $SO_{2,aq}$, 1,2-ISOPOOH contributes much more to atmospheric sulfate production than the 4,3 isomer (Dovrou et al., 2019b).

The differences between both the sulfate yield and the pH-dependence of rate constants suggest that the chemical mechanisms of $SO_{2,aq}$ oxidation via $H_2O_2$, and ISOPOOH are different. Hoffman and Edwards (1975) proposed a mechanism for the oxidation of $HSO_3^-$ by $H_2O_2$:

$$H_2O_2 + HSO_3^- \leftrightharpoons HOOSO_2^- + H_2O \qquad (R1)$$

$$HOOSO_2^- + H^+ \rightarrow HSO_4^- + H^+ \qquad (R2)$$

with reaction (R2), rearrangement of peroxysulfite, being the rate limiting step. To compare by $H_2O_2$, and understand the mechanism of $SO_{2,aq}$ oxidation with ISOPOOH, time-dependent ${}^1$H-NMR spectra of the reaction were obtained.





## 2 Materials and methods

### 2.1 Chemical and sample preparation

Sodium metabisulfite was purchased from Sigma Aldrich (purity ≥99%) and used as the source of bisulfite ($HSO_3^-$) in the solutions (Dovrou et al., 2019b). Formaldehyde solution (37 wt. % in $H_2O$, 10-15% methanol as stabilizer), methyl vinyl ketone (MVK, purity 99%), formaldehyde-sodium bisulfite adduct (hydroxymethanesulfonate, HMS, purity 95%), acetic acid (purity ≥99.7%), dimethyl sulfoxide (DMSO, purity ≥99.7%), 2-methyl-2-vinyloxirane (purity 95%) and 3-(trimethylsilyl)-
1-propanesulfonic acid sodium salt (purity 97%) were purchased from Sigma Aldrich. The two main ISOPOOH isomers, 1,2-ISOPOOH and 4,3-ISOPOOH, were synthesized in lab according to the procedures described by Rivera-Rios (2018). Hydrogen peroxide ($H_2O_2$) was purchased from Sigma Aldrich (30 wt. % in $H_2O$) and filtered Milli-Q water and deuterium oxide ($D_2O$) were used as solvents.

2-Methyl-3-butene-1,2-diol (1,2-ISOPOH) was synthesized, following the procedure in Zhang et al. (2012) and Bates et al. (2014), by hydrolysing 2-methyl-2-vinyloxirane. Approximately 5400 mg of 2-methyl-2-vinyloxirane was dissolved in 54 mL of 0.1 M HCl at 50°C for 30 min. The procedure was modified by saturating the reaction mixture with sodium chloride (NaCl), instead of lyophilizing it. Subsequently, the diol was isolated via extraction with diethyl ether. The diethyl ether solution was then evaporated under reduced pressure to isolate the diol. The purity of the diol was determined to be 70% using NMR
analysis, and the identified impurities were diethyl ether, water, and the 1,2-ISOPOH dimer.

Samples for kinetics experiments were prepared with a ratio of $\frac{[HSO_3^-]}{[hydroperoxide]} = \frac{2}{1}$ and $\frac{[HSO_3^-]}{[MVK]} = \frac{2}{1}$, as discussed in Dovrou et al. (2019b). The concentrations of the IC and NMR experiments were 10s of μM and a few mM, respectively, based on the sensitivity of the methods. For the NMR analysis of standards, 1 mM concentrations were used. After completion of reactions,
0.1-2 mM standards were added to identify proposed products by observing increases in the intensity of the corresponding peaks in the spectra.

### 2.2 Ion chromatography and NMR analysis

A Dionex ICS-5000+ Ion Chromatography (IC) system and an Agilent I500B NMR were used to analyze the samples and quantify sulfate produced. The AG12A guard column and the AS12A analytical column (Dionex Ionpac) were selected in
order to separate sulfur-containing species (Dovrou et al., 2019a). The mobile phase was 4.5 mM:1.4 mM sodium carbonate: sodium bicarbonate with flow rate 1.5 mL · $min^{-1}$ (Dovrou et al., 2019b). [1]H-NMR was used to identify the organic reaction products. 16 and 32 scan averages were chosen, with a relaxation delay 45 sec and at proton operating frequency 400 MHz.



All experiments were repeated at least four times, and each sample was prepared separately prior to analysis and temperature
and pH were monitored. DMSO, methanol and 3-(trimethylsilyl)-1-propanesulfonic acid sodium salt ($(CH_3)_3Si(CH_2)_3SO_3Na$)
were used as internal standards for quantitative yields.

## 2.3 GEOS-Chem simulations

GEOS-Chem simulations were performed identically to those described in Dovrou et al. (2019b). In brief, GEOS-Chem model
version 11-02d, which includes bromine chemistry relevant to $SO_2$ oxidation, was used as a base, and the chemical mechanism
was updated with the isoprene chemistry described in Bates and Jacob (2019). Emissions of isoprene were derived from the
MEGAN v2.1 inventory and scaled uniformly to $535\ Tg \cdot yr^{-1}$. The simulations performed are described in detail in the work
of Dovrou et al. (2019b). All simulations were performed at 4° latitude by 5° longitude horizontal resolution with 72 vertical
levels and spun up for one year, after which annual and summertime simulations were used for the results reported herein
under both modern (2016) and pre-industrial conditions, in which anthropogenic emissions are removed and the global
methane distribution is scaled down by 60%.

## 3 Results and discussion

### 3.1 Product identification

Fig.1 shows the NMR spectra of 1,2- and 4,3-ISOPOOH standards in $D_2O$ (broad peak at 4.7ppm) together with the NMR
spectra after reaction of ISOPOOH with a two-fold excess of $SO_{2,aq}$ at three different pH values. After completion of the $SO_{2,aq}$
+ISOPOOH reaction at pH=5.5 the ISOPOOH precursors are not observable and a number of high intensity [1]H-NMR signals
are observed. Signals with the same shifts are observed under more acidic conditions but with slightly lower intensity, while
some of the smaller peaks from the pH=5.5 spectra of both isomers are not observed in the pH=4.5 and 3 spectra. This is
consistent with the hypothesis that the reaction and product distribution have a pH dependence and suggests competition
between different reaction paths. These different paths may also explain the pH dependence of the observed sulfate yields.


Products were identified via comparison of [1]H-NMR spectra of standards as well as spiking $SO_{2,aq}$+ISOPOOH mixtures after
completion of reaction with standards of the proposed products. Methyl vinyl ketone (MVK), methacrolein (MACR) and
(hydrated) formaldehyde (HCHO), all major products of gas-phase oxidation, were identified by their [1]H-NMR chemical shifts
(St. Clair et al., 2016). Due to its higher importance, the 1,2-ISOPOOH reaction and products were examined in more detail.
For this reaction at pH=5.5 MVK (30% molar yield), hydrated formaldehyde ($CH_2(OH)_2$), HMS (30% molar yield), 1,2-
ISOPOH (22% molar yield) and acetic acid (5% molar yield), were the main products (Fig. 2A and Table 2). The products of
the $SO_{2,aq}$+4,3-ISOPOOH reaction included $CH_2(OH)_2$, 4,3-ISOPOH and HMS (Fig. 2B). In both reactions, $CH_2(OH)_2$ is
under the large $D_2O$ peak, $\delta \approx 4.74$ ppm, which prevents quantification (Section 3.2). 2-methyl-2-vinyloxirane was also used


as a standard, but it was not observed as a product (Fig. S1). The standard spectrum showed that 2-methyl-2-vinyloxirane

hydrolyzed to 1,2-ISOPOH under the experimental conditions, although not very rapidly. Thus, we cannot exclude that 2-methyl-2-vinyloxirane is formed and rapidly hydrolyzed to 1,2-ISOPOH, although timescale of the experiments should have allowed its observation. Some minor product signals could not be identified; for example, a signal in the $SO_{2,aq}$+1,2-ISOPOOH pH=5.5 spectrum at $\delta$=1.43 ppm (Fig.1) is an alkane based on its chemical shift; however, we were not able to identify the corresponding compound.

**3.2 Proposed reaction pathways**

It is instructive to consider the previously studied oxidation of $SO_{2,aq}$ with $H_2O_2$ (R1-R2) and methyl hydroperoxide ($CH_3OOH$), R3, and to lesser degree peroxyacetic acid ($CH_3C(O)OOH$), R4, (Lind et al., 1987).

$$HOSO_2^- + CH_3OOH \rightarrow 0.73(SO_4^{2-} + H^+ + CH_3OH) + 0.27(CH_3OSO_3^- + H_2O) \qquad (R3)$$

$$HOSO_2^- + CH_3C(O)OOH \rightarrow SO_4^{2-} + H^+ + CH_3COOH \qquad (R4)$$

Reaction 3 has two pathways oxidizing S(IV) to S(VI): a higher yield pathway, R3a, forming sulfate and methanol and a lower yield pathway, R3b, forming methyl bisulfite.

$$HOSO_2^- + CH_3COOH \rightarrow SO_4^{2-} + H^+ + CH_3OH \qquad (R3a)$$

$$HOSO_2^- + CH_3COOH \rightarrow CH_3OSO_3^- + H_2O \qquad (R3b)$$

Lind et al.'s (1987) results for $CH_3OOH$ and $CH_3(O)OOH$ are consistent with a fast initial equilibrium, like R1, followed by

rearrangement to the products. In addition, Lind et al. (1987) stated that at pH values above 4.8 the $CH_3(O)OOH$ reaction had another mechanism, but the authors did not elaborate, only stated that the mechanism was first-order in both reactants. R3a and R4 are effectively oxygen atom transfer reactions, analogous to the net R1-R2 reactions of $H_2O_2$. R3b forms an organosulfate as the stable reaction product (Lind et al., 1987). In our discussion of ISOPOOH reaction pathways we focus on the atmospherically more important 1,2-ISOPOOH. Based on the product distribution we propose a non-fragmentation

pathway for 1,2-ISOPOOH forming 1,2-ISOPOH (22% molar yield at pH=5.5) and two fragmentation pathways producing MVK (30% molar yield at pH=5.5) together with formaldehyde/HMS (>30% molar yield at pH=5.5) and acetic acid (5% molar yield at pH=5.5) with other small species such as CO. These pathways accounting for 57% of 1,2-ISOPOOH reaction at pH=5.5 will be discussed below.

**3.2.1 Non-fragmentation pathway**

Analogous to R1, we also consider an initial fast equilibrium of ISOPOOH with $SO_{2,aq}$. However, because of the lower symmetry than $H_2O_2$, there are two possibilities, depending on which peroxy O is involved in the nucleophilic attack on sulfur.



(R5a)




(R5b)

Peroxysulfite from R5a will rearrange to sulfate following R2. The organo peroxysulfite from R5b could potentially rearrange to form an organosulfate (R6a) which is known to rapidly hydrolyze forming sulfate and ISOPOH (Darer et al., 2011; Hu et al., 2011). This rapid hydrolysis of tertiary sulfate likely also applies to the organo peroxysulfite. The hydrolysis product (R6b) is again ISOPOH and peroxysulfite, which will rearrange to sulfate following R2.


(R6a)

(R6b)

The net outcome of all pathways is effectively an oxygen atom transfer from 1,2-ISOPOOH to sulfite to form sulfate and 1,2-ISOPOH. This net mechanism is consistent with the NMR spectra and IC analysis which show clear evidence for ISOPOH and sulfate production. This effective reaction path is shown in the left-hand side of Fig. 3[1]. This pathway accounts for 22%

of 1,2-ISOPOOH reaction at pH=5.5.

### 3.2.2 Fragmentation pathways

1,2-ISOPOH only has about 2/3 of the yield of MVK at all pH values which is strong evidence that an additional pathway, likely not present for $CH_3OOH$, exists. MVK and formaldehyde, either as its hydrate or as the adduct with the excess $SO_{2,aq}$, are in fact the highest yield products. These products have to involve breaking of the carbon backbone, which strongly suggests

a radical mechanism. The products are identical to those resulting from oxidation of isoprene by OH under high-NO conditions, which involves an alkoxy radical (1,2-ISOPO, Fig. 3). In addition, the products are identical to those observed for gas-phase ISOPOOH reacting on metal surfaces (Rivera-Rios et al., 2014). For the latter process, Rivera-Rios et al. (2014) proposed cleavage of the 1,2-ISOPOOH peroxy bond producing 1,2-ISOPO which rapidly forms MVK and formaldehyde. The high sulfate yield at all pHs implies that both the ISOPOH and the fragmentation channel have to produce sulfate. In combination

with the previous results mentioned above, we propose that the second pathway involves cleavage of the peroxy bond of the organo peroxysulfite resulting in 1,2-ISOPO, which rapidly forms MVK/formaldehyde, and a sulfite radical. The sulfite radical rapidly reacts with $O_2$ producing a peroxysulfate radical, which can undergo further reaction, e.g., with $SO_{2,aq}$ or organic molecules to form sulfate and oxidized organics (Neta and Huie, 1985; Yang et al., 2015). The subsequent (secondary) reaction of HCHO with $SO_{2,aq}$ explains formation of HMS and thus the pathway forms MVK, HCHO and the sulfite radical as first

generation products with the sulfite radical likely forming sulfate subsequently (Fig. 3) (Dovrou et al., 2019a; Munger et al., 1984; Munger et al., 1986). This pathway accounts for 30% of 1,2-ISOPOOH reaction at pH=5.5.



The observation of acetic acid also requires a fragmentation of the backbone. This pathway likely involves formation of additional formaldehyde and CO, which are both not quantifiable in our experiments. For this reason, this is the most uncertain pathway although it is clear that it accounts for 5% of 1,2-ISOPOOH reaction at pH=5.5, which makes the sum of the non-fragmentation and the two fragmentation pathways 57% at pH=5.5 (Rivera-Rios et al., 2014; Paulot et al., 2009).

There likely are three contributors to the carbon yield of less than 100%. Hydrated formaldehyde could not be quantified due to interference with $D_2O$ although this likely is already accounted for in the MVK and acetic acid pathways. However, MVK could also repartition to the gas-phase, as discussed in section 4; thus, its aqueous contribution might be underestimated. It is likely that formation of CO or other small molecules that potentially partition to the gas-phase are not detected by NMR. Lastly, it is also possible that numerous small-yield products are formed that are below the NMR detection limit.

### 3.3 pH dependence

The sulfate yield increases while the observed carbon decreases with decreasing pH, and the ratio of ISOPOH to MVK is pH independent. Moreover, at lower pH, ISOPOH and MVK molar yields are reduced by no more than 10%, and the acetic acid path is greatly reduced. We propose two explanations for the fact that the observed carbon balance decreases with decreasing pH. Kok et al. (1986) showed that the formation of HMS is a factor of 10 slower at pH=4 than pH=5, which could explain that at pH≤4.5 the sulfate yield observed in the oxidation of $SO_{2,aq}$ with ISOPOOH is 100%, indicating that $SO_{2,aq}$ is oxidized faster to sulfate than reacting with formaldehyde to form HMS, while pH=5.5 the sulfate yield is lower. Thus, the decreased HMS yield is attributed to slow HMS formation so that more formaldehyde stays in the hydrated form which could not be quantified. Second, at lower pH acetic acid may partition to the gas-phase and thus it cannot be observed. The results do not provide a clear explanation for the lower sulfate yield at pH=5.5. The non-fragmentation (ISOPOH) pathway, which is more important at pH=5.5, cannot explain this as it should have a 100% sulfate yield. Thus, the cause has to result from one of the fragmentation pathways. It is possible that at higher pH the sulfite radical has a lower yield of forming sulfate, however, studying this goes beyond the scope of this work.

In summary, we observe clear evidence for a non-fragmentation pathway that likely is similar to the mechanism of $H_2O_2$ and $CH_3OOH$. A difference to $CH_3OOH$ is that in contrast to methyl bisulfate the organo sulfates from ISOPOOH hydrolyze easily, especially the tertiary one from 1,2-ISOPOOH. The non-fragmentation pathway effectively corresponds to an oxygen atom transfer and in contrast to $H_2O_2$ and $CH_3OOH$ this is not the major pathway. We observe fragmentation pathways that we hypothesize involve cleavage of the peroxy bond based on previous work on ISOPOOH. Such a pathway was not observed for $CH_3OOH$ as the product of this channel would be neither methanol nor methyl bisulfate. There are at least two fragmentation pathways, with the major observed one producing MVK and HCHO, which subsequently forms HMS, and a minor one forming acetic acid and probably small molecules including CO. It is likely that additional smaller molecules that partition to the gas-



phase are formed as well as other molecules at low concentration that cannot be detected via NMR, which could include organo-sulfur compounds from the reaction of the sulfite radical with organics.

## 4 Conclusions and atmospheric implications

This work is primarily useful as a foundation for the mechanistic understanding of the reactivity of multifunctional organic hydroperoxides with $SO_{2,aq}$. It shows that they have a significantly more complex behaviour with channels analogous to $H_2O_2$

and $CH_3OOH$ but also fragmentation channels. Understanding the mechanisms of these oxidation reactions can help clarify the composition of cloud droplet residual aerosols and adds new pathways to the isoprene oxidation mechanism.

In the atmosphere, MVK and MACR will partition to the gas-phase as they have high vapor pressures of 11 and 16 kPa and low Henry's law constants of 41 and 6.5 $M \cdot atm^{-1}$, respectively (Iraci et al., 1999). The diffusion time of MVK in a cloud

droplet is less than 25 seconds. The rate of the reaction of MVK with $SO_{2,aq}$ was estimated using NMR analysis, from which we estimate a lower limit for the rate constant of $k \geq 10^4 \, M^{-1} \cdot s^{-1}$. However, considering the diffusion time, $SO_{2,aq}$ and expected MVK concentrations, the formation of the organosulfur compound by reaction of MVK with $SO_{2,aq}$ is unlikely in the atmosphere.

The annual production of the products formed via the reaction of ISOPOOH with $SO_{2,aq}$ in the atmosphere was calculated using the estimated ISOPOOH amount participating in this reaction from GEOS-Chem (Jacob et al., 2001; Park et al., 2004; Alexander et al., 2012; Dovrou et al., 2019b). Globally, it is estimated that, under current conditions, 4.2 $Tg \cdot yr^{-1}$ of gas-phase 1,2-ISOPOOH reacts with $SO_{2,aq}$, representing 3.2 % of the global loss of the ISOPOOH isomer and 1.7% of the global production of sulfate (Dovrou et al., 2019b). The 1,2-ISOPOOH pathway contributes to the production of 0.40-0.53, 0.02-

0.06, 0.28-0.36, 0.95-1.31 and 1.40-1.52 $Tg \cdot yr^{-1}$ of MVK, acetic acid, HCHO, HMS and 1,2-ISOPOH, respectively (ranges are for the differences in yields across the experimental pH range). The production of MVK from these ISOPOOH+$SO_{2,aq}$ oxidation pathways is equivalent to 0.5% of the total global production under present conditions. Under preindustrial conditions, the total atmospheric burden of $SO_2$ decreases; thus, the amount of ISOPOOH reacting with $SO_{2,aq}$ decreases somewhat: 2.4 $Tg \cdot yr^{-1}$ of gas-phase 1,2-ISOPOOH react with $SO_{2,aq}$, representing 1.6% of the global respective losses of the

isomer and 1.0% of the global production of sulfate. The reaction of 1,2-ISOPOOH+ $SO_{2,aq}$ results in 0.23-0.30, 0.01-0.04, 0.16-0.21, 0.55-0.76 and 0.80-0.88 $Tg \cdot yr^{-1}$ of MVK, acetic acid, HCHO, HMS and 1,2-ISOPOH, respectively. The production of MVK from these ISOPOOH+ $SO_{2,aq}$ oxidation pathways is equivalent to 3.7% of the total global production under pre-industrial conditions. These results show that the importance of these pathways lies in the production of sulfate on regional scales, especially for isoprene-dominated regions, rather than the formation of the organic reaction products. The

conversion of bisulfite to sulfate via the 1,2-ISOPOOH pathway can contribute up to ~50% of total sulfate production in pristine regions such as the Amazon, and ~3% in more polluted regions such as the Southeast United States under current atmospheric conditions (Dovrou et al., 2019b). The Southeast United States is an area of interest as $NO_x$ levels are decreasing,





creating an atmosphere where hydroperoxide formation can be favored, which may increase their contribution to sulfate formation; in a simulation without anthropogenic emissions, the contribution of ISOPOOH to sulfate formation increases from
~3% to ~20% (Dovrou et al., 2019b).

*Data availability*. The data used in this work are available upon request. Please email Eleni Dovrou (dovrouel@gmail.com)

*Supplement*. Supplemental information is available at xx.

*Author contributions*. E.D. conducted the ion chromatography experiments and E.D. and J.C.R.R. conducted the NMR experiments. E.D., K.H.B. and F.N.K. analysed the data. J.C.R.R.  synthesized the two ISOPOOH isomers. J.L.C. and J.D.S. synthesized the 1,2-ISOPOH to provide the [1]H-NMR standard, and J.L.C. provided the 2-methyl-2-vinyloxirane and acetic acid [1]H-NMR standards. E.D. prepared the paper with contributions from K.H.B., J.C.R.R., J.L.C., J.D.S. and F.N.K.

*Competing interests*. The authors declare no competing financial interest.

*Acknowledgements*. This work was supported by the Harvard Global Institute. Eleni Dovrou is grateful for the Onassis Foundation scholarship for Hellenes.

*Financial support*. This research was supported by the Harvard Global Institute.

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



**Figure 1: Proton NMR spectra ($^1$H-NMR, 400MHz, $D_2O$) of (A) 1,2-ISOPOOH (d) and the products of the reaction of $SO_{2,aq}$+1,2-ISOPOOH at (a) pH=3, (b) pH=4.5 and (c) pH=5.5 and (B) 4,3-ISOPOOH (d) and the products of the reactions of $SO_{2,aq}$+4,3-ISOPOOH at (a) pH=3, (b) pH=4.5 and (c) pH=5.5. The concentration of the ISOPOOH isomers was 1 mM and the concentration of $SO_{2,aq}$, was 2 mM. The spectra are focused on the areas of change. Chemical structures of (A,d) 1,2-ISOPOOH and (B,d) 4,3-ISOPOOH are presented . The labels at each peak represent the hydrogens of the compounds and the spectra are shifted for $D_2O$ at 4.67 ppm.**






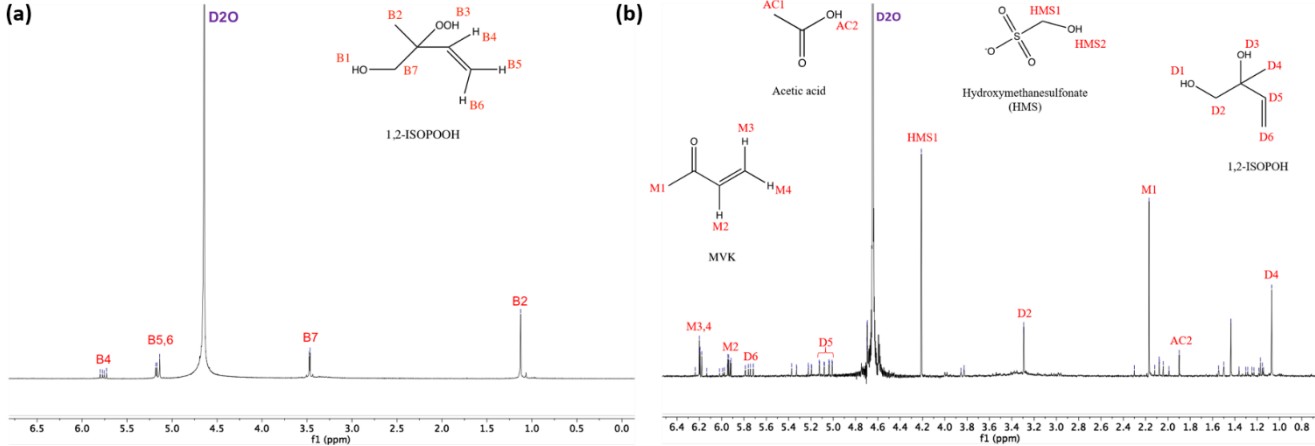

**Figure 2. Proton NMR spectra (1H-NMR, 400MHz, D₂O) for 1,2-ISOPOOH (a), and products of the reaction of SO$_{2,aq}$+1,2-ISOPOOH (b). The marked peaks represent the position of the hydrogens for each molecule. The pH of the samples was 5.5 and the concentrations used were [1,2-ISOPOOH]=1 mM and [SO$_{2,aq}$]=2 mM. D₂O shift at 4.67 ppm.**




**Figure 3. Proposed chemical mechanisms of the oxidation of $SO_{2,aq}$ by a) 1,2-ISOPOOH and b) 4,3-ISOPOOH. There are two competing mechanisms: after ISOPOOH reacts with $SO_{2,aq}$, displacing water, a hydrolysis reaction is taking place [1] or an O-O bond breakage [2]. In mechanism [1], the product hydrolysis results in the same intermediate that the reaction of $SO_{2,aq}$ with $H_2O_2$ is forming and either a formation of a diol or an epoxide is being generated (Figure S2). In mechanism [2], an alkoxy radical and sulfite radical are formed leading to the production of MVK, MACR, HCHO and other products.**






**Table 1. Sulfate production rate constants and yields in the aqueous phase due to oxidation of $SO_{2,aq}$ via 1,2-ISOPOOH and 4,3-ISOPOOH.(Dovrou et al., 2019b)**

|  | 1,2-ISOPOOH+$HSO_3^-$ | | 4,3-ISOPOOH+$HSO_3^-$ | |
|---|---|---|---|---|
|  | Rate constant ($M^{-1} s^{-1}$) | Sulfate yield (%) | Rate constant ($M^{-1} s^{-1}$) | Sulfate yield (%) |
| pH=5.5 | $1.65(\pm0.28) \cdot 10^3$ | 67 | $1.80(\pm0.23) \cdot 10^2$ | 83 |
| pH=4.5 | $1.00(\pm0.28) \cdot 10^3$ | 100 | $1.90(\pm0.18) \cdot 10^2$ | 100 |
| pH=3 | $1.00(\pm0.42) \cdot 10^3$ | 100 | $2.90(\pm0.30) \cdot 10^2$ | 100 |

**Table 2. Carbon budget of the 1,2-ISOPOOH+$SO_{2,aq}$ reaction and molar yields. Carbon concentrations are in mM. (Hydrated) formaldehyde could not be quantified as it is under the large $D_2O$ signal. At lower pH, acetic acid may be lost to the gas-phase.**

|  | Carbon Concentration | | | Molar Yield | | |
|---|---|---|---|---|---|---|
|  | pH=5.5 | pH=4.5 | pH=3 | pH=5.5 | pH=4.5 | pH=3 |
| 1,2-ISOPOOH | 5 | 5 | 5 |  |  |  |
| HMS | 0.3 | 0.2 | 0.2 | 30($\pm$4)% | 20($\pm$3)% | 20($\pm$5)% |
| MVK | 1.2 | 0.8 | 0.9 | 30($\pm$6)% | 20($\pm$4)% | 23($\pm$4)% |
| 1,2-ISOPOH | 1.1 | 0.7 | 0.8 | 22($\pm$4)% | 14($\pm$2)% | 16($\pm$6)% |
| Acetic acid | 0.1 | 0.01 | 0.02 | 5($\pm$1)% | 0.5($\pm$0.03)% | 1($\pm$0.08)% |
| Sum | 2.7 | 1.7 | 1.9 | 57% | 35% | 40% |