# Peer review of "Towards a Chemical Mechanism of the Oxidation of Aqueous Sulfur Dioxide via Isoprene Hydroxyl Hydroperoxides (ISOPOOH)"

_Atmospheric Chemistry and Physics, 2021_

## Author Comment (AC1)

Reply to comments of Referee Becky Alexander for the manuscript "Towards a Chemical Mechanism of the Oxidation of Aqueous Sulfur Dioxide via Isoprene Hydroxyl Hydroperoxides (ISOPOOH)"

We would like to thank the referee for the comments, that helped improve our manuscript. The comments are mentioned below with italic followed by our responses.

*Significant issue #1: SO2 dissolves in cloud water to form SO2H2O, bisulfite (HSO3-) and sulfite (SO32-). The authors refer to SO2,aq throughout the text, but never refer to specific species in the aqueous phase. The relative importance of each species is pH dependent, and since they are evaluating the pH dependence of the reactions, this seems important. The mechanisms proposed all involve reactions with bisulfite, with sulfite not mentioned, even for the higher pH values where sulfite concentration will be significant. They also show that the sulfate product yield is lower at the higher pH values, and I wonder if this has to do with less of the dissolved SO2 being present as bisulfite, and more present as sulfite. Does the reaction of ISOPOOH not occur with sulfite? This should be discussed.*

We agree with the referee that the speciation of $SO_{2,aq}$ should be clearly mentioned and discussed in the manuscript. Therefore, we clarify that in the examined pH range of 3-6 the dominant form of $SO_{2,aq}$ is bisulfite ($HSO_3^-$) by including the following statement and figure (Fig. S1).

We add the following statement in line 47 page 2:

"$SO_{2,aq}$ reacts with water to form sulfurous acid ($SO_2 \cdot H_2O$), which dissociates to bisulfite ($HSO_3^-$) when pH>2. At higher pH (pH > 6), $HSO_3^-$ subsequently dissociates to form sulfite ($SO_3^{2-}$). In cloud pH range of 3-6 the dominant form of $SO_{2,aq}$ is $HSO_3^-$ and our study investigated pH values of 3, 4.5 and 5.5 (Fig. S1)."

All the experiments were performed with bisulfite as the dominant S(IV) species. For the pH range examined, 3-6, bisulfite is the dominant sulfur species as shown in Figure S1. Thus, we do not expect sulfite to be important in this case. At the highest pH value of our experiments, pH=5.5, the ratio of bisulfite to sulfite is approximately 60. However, if the rate constant of ISOPOOH with sulfite is orders of magnitude faster than the reaction of ISOPOOH with bisulfite, and the reaction mechanism is different for sulfite than bisulfite, this could indeed contribute to changes in product distributions. Our experiments do not allow unambiguous and quantitative analysis of this aspect. Performing experiments at higher pH to shift the equilibrium to sulfite has the significant complication that formaldehyde will react rapidly with it to form HMS. However, even if sulfite is important, our results are useful for model simulations as they capture the effective reactivity at each pH value examined, which covers the cloud droplet relevant range, i.e., the way the reaction occurs at the specific pH in clouds, even if at the high pH, so not cloud relevant, there would be a contribution from sulfite.

We add the following statement in line 206 and page 7:

"In addition, S(IV) speciation might affect the sulfate yield at high pH. While the dominant form of $SO_{2,aq}$ in our experiments is bisulfite, at the highest pH value of our experiments, pH=5.5, sulfite is likely formed and could potentially react with ISOPOOH (Fig. S1). The ratio of bisulfite to sulfite at pH=5.5 is approximately 60. However, if the rate constant of ISOPOOH with sulfite is orders of magnitude faster than the reaction of ISOPOOH with bisulfite, and the reaction mechanism is different for sulfite than bisulfite, this could contribute to changes in product distributions. Our experiments do not allow unambiguous and quantitative analysis of this aspect. Performing experiments at higher pH, exceeding the cloud relevant pH, to shift the equilibrium to sulfite has the significant complication that formaldehyde will react rapidly with it to form HMS."

[Figure]

**Figure S1. Mole fraction concentrations of S(IV) species vs pH. The green shaded area shows the pH range of 3-6 and the three pH values examined in the present work: pH=3 (crimson), pH=4.5 (purple) and pH=5.5 (dark yellow). The dominant form of $SO_{2,aq}$ under these conditions is bisulfite ($HSO_3^-$) (Seinfeld and Pandis, 2016).**

*Significant issue #2: I'm confused by the modeling. Section 2.3 describes the model simulation (though I would not consider the description of the model simulations to be complete). The model results are not discussed until the conclusion, in the very last paragraph of the paper. This is not an appropriate place to present results for the first time. The conclusion section refers to Dovrou et al. (2019), suggesting that no new model simulations were performed for this paper. If no new model simulations were performed, then the existence of section 2.3 is misleading. If indeed no new model simulations were performed, then I don't see the point of discussing model results if the results of the laboratory experiments were not included in the model for the present study. If the model was run with the new information gleaned from the presented laboratory studies, then model results (with figures) should be shown in the results section. In sum, I have no idea what to*

*make of the model results discussed in the last paragraph of the paper or if the model includes the new information learned from the laboratory studies.*

The focus of this work is to investigate the organic carbon contribution of the reaction of ISOPOOH with bisulfite. Thus, we re-analyzed the results from the simulations conducted in the work of Dovrou et al. (2019b). We will clarify that in the manuscript by removing the Section 2.3, as suggested, and add the following information on Section 4. Section 4 includes atmospheric implications thus we chose to discuss them together with the conclusions to highlight the importance of sulfate formation in global and regional scales, present how the atmosphere is affected by the organic products formed via the examined reactions and avoid repetition. However, in order to present the atmospheric implications clearly, we will include the following statement and lines 223-225 as a separate section, as suggested.

We add the following information in line 231 page 8:

"The results from the simulations performed in the work of Dovrou et al. (2019b) were re-analyzed with respect to the amount of ISOPOOH reacting in each pathway. Dovrou et al. (2019b) did not consider the organic products of the ISOPOOH+$SO_{2,aq}$ reaction as they focused on an estimate of the globally and regionally produced sulfate (see Dovrou et al. (2019b) for all details on model configurations). This work focused on the organic reaction products, primarily to provide the first insight into this for multifunctional organic hydroperoxides, and to evaluate whether in addition to the clear regional importance for sulfate, the reactions are important for the organic carbon budget. To be specific, the amount of ISOPOOH reacting in each pathway was calculated from the simulations and was adjusted for each organic product using the product yields. The results were scaled by the relative molecular weights in order to calculate the total production of the compounds of interest."

In addition, the conclusion will include the following to address this comments:

"This work is primarily useful as a foundation for the mechanistic understanding of the reactivity of multifunctional organic hydroperoxides with $SO_{2,aq}$, for which there is no previous data in the literature. The results show that multifunctional organic hydroperoxides have a significantly more complex behaviour with channels analogous to $H_2O_2$ and $CH_3OOH$ but also fragmentation channels. Dovrou et al. (2019b) demonstrated the importance of these pathways in the production of sulfate under isoprene-dominated regions compared to the formation of the organic reaction products. Our work shows that, although the reactions do not play an important role for the organic carbon budget, it advances understanding of the atmospheric chemistry of multifunctional organic hydroperoxides, and adds new pathways to the isoprene oxidation mechanism. Furthermore, the results can help clarify the composition of cloud droplet residual aerosols."

*Minor issues:*

*Line 91: Should this be referencing Dovrou et al. (2019a) (not b)?*

The work of Dovrou et al. (2019a) used mobile phase of 4.5mM: 0.8mMsodium carbonate to sodium bicarbonate with a flow rate of 1mL/min, while in the work of Dovrou et al. (2019b) mobile phase of 4.5 mM:1.4 mM sodium carbonate: sodium bicarbonate with flow rate 1.5mL/min was used in order to achieve higher peak resolution. In the present work we also used the mobile phase and flow rate used in the study of Dovrou et al. (2019b).

*Section 2.3: Model needs a doi. You included a reference that describes the isoprene chemistry, but you also need to provide a reference that describes the sulfur chemistry in the model. You need to state which met fields were used (MERRA2? GEOS-FP?). What cloud pH calculation is being used? This seems relevant and should be referenced. Provide a reference for how bromine chemistry impacts SO2 oxidation (Chen et al, 2017), or delete this line as I'm not sure how it's directly relevant to this work.*

As suggested by the referee in a previous comment (issue #2) we will remove Section 2.3 and reference the work of Dovrou et al. (2019b) for detailed information regarding the model used in this work.

*Reaction R3 has CH3OOH as a reactant, while in R3a and R3b its CH3COOH (there's an extra C in 3a and 3b)*

We would like to thank the referee for pointing out the typo in reactions R3a and R3b. We corrected the reactant to CH3OOH.

Reference:

Seinfeld, J. H. and Pandis, S. N. (Eds): Atmospheric Chemistry and Physics: From Air Pollution to Climate Change, John Wiley & Sons Publications, Hoboken, New Jersey, 2016

---

## Author Comment (AC2)

Reply to comments of Anonymous Referee for the manuscript "Towards a Chemical Mechanism of the Oxidation of Aqueous Sulfur Dioxide via Isoprene Hydroxyl Hydroperoxides (ISOPOOH)"

We would like to thank the referee for the comments, that helped the revise and improve our manuscript. The comments are mentioned below with italic followed by our responses.

*Specific comments*

*Line 60: For the benefit of readers with less knowledge of the sulfur dioxide oxidation mechanism, please include reactions that include the accommodation and dissolution of gaseous sulfur dioxide and its conversion to $HSO_3^-$.*

We would like to thank the referee for the recommendation. As suggested we added the following statement in the main text and the following information as a new section in the Supplement.

We add the following statement in line 47 page 2:

"$SO_{2,aq}$ reacts with water to form sulfurous acid ($SO_2 \cdot H_2O$), which dissociates to bisulfite ($HSO_3^-$) when pH>2. At higher pH (pH > 6), $HSO_3^-$ subsequently dissociates to form sulfite ($SO_3^{2-}$) (Seinfeld and Pandis, 2016). In the cloud pH range of 3-6 the dominant form of $SO_{2,aq}$ is $HSO_3^-$ and our study investigated pH values of 3, 4.5 and 5.5 (Fig. S1)."

We add in Supplemental information:

**1. Sulfur dioxide**

Sulfur dioxide ($SO_2$) has a Henry's law constant of $H_{SO_2}$=1.3 M atm$^{-1}$ and mass accommodation coefficient of $\gamma_{SO_2}$=0.11, which does not significantly change with temperature (Worsnop et al., 1989). Thus, $SO_2$ can dissolve ($SO_{2,aq}$) into cloud and fog water, where it can be rapidly oxidized to form sulfate.(Lind et al., 1987; Hegg and Hobbs, 1982; Shen et al., 2012; Harris et al., 2014; Dovrou et al., 2019b) $SO_{2,aq}$ reacts with water to form bisulfite ($HSO_3^-$) at pH$\gtrsim$2, which dissociates to form sulfite ($SO_3^{2-}$) at pH$\gtrsim$6 (RS1-RS3, Fig. S1) (Seinfeld and Pandis, 2016).

$$SO_{2,g} + H_2O \rightleftharpoons SO_2 \cdot H_2O \qquad \text{(RS1)}$$

$$SO_2 \cdot H_2O \rightleftharpoons H^+ + HSO_3^- \qquad \text{(RS2)}$$

$$HSO_3^- \rightleftharpoons H^+ + SO_3^{2-} \qquad \text{(RS3)"}$$

[Figure]

**Figure S1. Mole fraction concentrations of S(IV) species vs pH. The green shaded area shows the pH range of 3-6 and the three pH values examined in the present work: pH=3 (crimson), pH=4.5 (purple) and pH=5.5 (dark yellow). The dominant form of SO$_{2,aq}$ under these conditions is bisulfite (HSO$_3^-$) (Seinfeld and Pandis, 2016).**

*Line 92: I'm puzzled by the need for the extremely long relaxation delays (45 seconds as compared to the usual 1 second for quantitative 1H NMR). I realize that there is some discussion of this problem in the SI, but I'm not quite sure I understand how the information reported establishes that the inconsistency in the integrations at different pH values is due to differences in the relaxation time for different 1H nuclei. Wouldn't this be better established by reporting the integrations (at pH = 3.0, where the inconsistencies are the greatest) for different relaxation times? I assume that the authors did this kind of a study, but I don't see it reported anywhere in the manuscript. Given what is reported in the manuscript, it seems to me that an alternative explanation for the data presented in Table S1 is simply that some species are either more volatile, less aqueous soluble, or consumed by secondary reactions at lower pHs and/or with different standards dissolved in solution.*

We would like to thank the referee for the comments. Regarding the long relaxation delays, we did experimental runs varying that parameter, and 45 seconds were chosen in order to achieve optimal resolution in combination with the relatively quick experimental runs to monitor the reactions. Shorter relaxation delays were generating noise, thus unreliable spectra. All experiments were conducted under the same conditions and the only changes among same pH runs were the relaxation delay prior to setting it to 45 seconds.

The explanation provided addresses the difference between Figure S3 and Table S1 for the peak intensities. Regarding the data presented in Table S1, they are all conducted under the same conditions per pH value using two different standards. The relative ratios of the integrated peaks

to the standard are similar for each standard, considering also that 0.5 mM of 3-(trimethylsilyl)-1-propanesulfonic acid sodium salt ($(CH_3)_3Si(CH_2)_3SO_3Na$) and 0.1 mM DMSO were used. Thus, the difference in the integration values presented are due to the different concentration used in each standard. In addition, we do not have any evidence indicating interaction of the products with the standards used.

*Line 124: While it might be difficult to detect a 1.2 mM (the concentration of the coproduct MVK at pH = 5.5) species with 13C NMR, the hydrated formaldehyde species could, in principle, be detected and quantified (again with long relaxation delays). In general, 13C NMR could have been useful for looking for other species that might have been overlapped in the 1H spectrum in the effort to understand why a significant amount of carbon was not quantified. Did the authors consider this approach?*

We performed experiments using 13C NMR and also 2D C and H NMR and the results did not provide additional information. To be more specific, we used relaxation times of 45 seconds and longer and the hydrated formaldehyde was not able to be quantified. We also tried to use 13C NMR for additional quantification of species but, unfortunately, the spectra were not clear enough compared to the 1H NMR spectra. Thus, we chose to continue and complete the analysis using 1H NMR.

*Line 128: The unidentified peak at 1.43 ppm might be more appropriately referred to as a methyl group not adjacent to a carbonyl group rather than as an "alkane."*

Due to the uncertainty of the identification of the peak and the fact that an alkane typically has a 1H NMR shift of ~0.8-1.9 we chose an alkane as a possible explanation. However, we rephrased the sentence, as suggested, as:

"…at δ=1.43 ppm (Fig.1) is likely a methyl group not adjacent to a carbonyl group based on its chemical shift,…"

*Line 167: I'm confused about what "2/3" refers to here. Isn't 30% of 1,2-ISOPOOH ending up as MVK?*

We clarified that statement, as it is referred to the products of 1,2-ISOPOOH, 1,2-ISOPOH and MVK,:

"1,2-ISOPOH yield is only about 2/3 of the yield of MVK at all pH values…"

*Technical comments*

*Line 83: The abbreviation IC precedes the definition of the acronym on line 88.*

We would like to thank the referee. We added this information on line 82:

"The concentrations of the ion chromatography (IC) and nuclear magnetic resonance (NMR) spectrometry experiments…"

*Line 108: Since the experiments use HSO3- as the reactant (and don't add actually add gaseous SO2 to an aqueous solution), it would be more clear if HSO3- is specifically identified as the reactant in the discussion of the experiments. This would be made more clear if the full mechanism were given in the introduction as suggested in my comment above.*

We added a clarifying statement, as mentioned above, in line 47 page 2 and we mention that the dominant form of $SO_{2,aq}$, in the experiments conducted in this work, was bisulfite.

*Figure 3: It would help make the figures more "stand alone" if they also included the yield information given in Table 2.*

We appreciate this comment, it is a very good suggestion. We add a new figure in the main text (Fig. 4) where the mechanism of 1,2-ISOPOOH+ $SO_{2,aq}$ at pH=5.5 is presented with the yields.

**Figure 4. Proposed chemical mechanisms of the oxidation of $SO_{2,aq}$ by 1,2-ISOPOOH with carbon yields of identified products at pH=5.5 (Table 2) presented in purple. There are two competing mechanisms: after ISOPOOH reacts with $SO_{2,aq}$, displacing water, a hydrolysis reaction is taking place [1] or an O-O bond breakage [2]. In mechanism [1], the product hydrolysis results in the same intermediate that the reaction of $SO_{2,aq}$ with H2O2 is forming and either a formation of a diol or an epoxide is being generated (Figure S2). In mechanism [2], an alkoxy radical and sulfite radical are formed**

**leading to the production of MVK, MACR, HCHO and other products. The sum of the products' carbon yields is 57% and the remaining 43% is attributed to hydrated formaldehyde, repartition of MVK to the gas-phase and possible formation of CO or other small molecules that have repartitioned to the gas-phase, as discussed in Section 3.2.2.**

References

Dovrou, E., Rivera-Rios, J. C., Bates, K. H. and Keutsch, F. N.: Sulfate Formation via Cloud Processing from Isoprene Hydroxyl Hydroperoxides (ISOPOOH), Environ. Sci. Technol., 12476 −12484, doi:10.1021/acs.est.9b04645, 2019b.

Harris, E., Sinha, B., Pinxteren, D. Van, Schneider, J., Poulain, L., Collett, J., Anna, B. D. and Fahlbusch, B.: In-cloud sulfate addition to single particles resolved with sulfur isotope analysis during HCCT-2010, Atmos. Chem. Phys., 4219–4235, doi:10.5194/acp-14-4219-2014, 2014.

Hegg, D. A. and Hobbs, P. V: Measurements of sulfate production in natural clouds, Atmos. Environ., 16(11), 2663–2668, doi:10.1016/0004-6981(82)90348-1, 1982.

Lind, J. A., Lazrus, A. L. and Kok, G. L.: Aqueous Phase Oxidation of Sulfur(Iv) By Hydrogen-Peroxide, Methylhydroperoxide, and Peroxyacetic Acid, J. Geophys. Res., 92(D4), 4171–4177, doi:10.1029/JD092iD04p04171, 1987.

Seinfeld, J. H. and Pandis, S. N. (Eds): Atmospheric Chemistry and Physics: From Air Pollution to Climate Change, John Wiley & Sons Publications, Hoboken, New Jersey, 2016

Shen, X., Lee, T., Guo, J., Wang, X., Li, P., Xu, P., Wang, Y., Ren, Y., Wang, W., Wang, T., Li, Y., Carn, S. A. and Collett, J. L.: Aqueous phase sulfate production in clouds in eastern China, Atmos. Environ., 62, 502–511, doi:10.1016/j.atmosenv.2012.07.079, 2012.

Worsnop, D. R., Zahniser, M. S., Kolb, C. E., Gardner, J. A., Watson, L. R., Van Doren, J. M., Jayne, J. T. and Davidovits, P.: Temperature dependence of mass accommodation of SO2 and H2O2 on aqueous surfaces, J. Phys. Chem., 93(3), 1159–1172, doi:10.1021/j100340a027, 1989.